# Increased Heparanase Levels in Urine during Acute Puumala Orthohantavirus Infection Are Associated with Disease Severity

**DOI:** 10.3390/v14030450

**Published:** 2022-02-22

**Authors:** Luz E. Cabrera, Constanze Schmotz, Moin A. Saleem, Sanna Lehtonen, Olli Vapalahti, Antti Vaheri, Satu Mäkelä, Jukka Mustonen, Tomas Strandin

**Affiliations:** 1Department of Virology, Medicum, University of Helsinki, 00290 Helsinki, Finland; olli.vapalahti@helsinki.fi (O.V.); antti.vaheri@helsinki.fi (A.V.); tomas.strandin@helsinki.fi (T.S.); 2Research Program for Clinical and Molecular Metabolism, University of Helsinki, 00290 Helsinki, Finland; constanze.schmotz@helsinki.fi; 3Bristol Renal, Translational Health Sciences, Bristol Medical School, University of Bristol, Bristol BS1 3NY, UK; m.saleem@bristol.ac.uk; 4Department of Pathology, University of Helsinki, 00290 Helsinki, Finland; sanna.h.lehtonen@helsinki.fi; 5Virology and Immunology, Diagnostic Center, Helsinki University Hospital (HUSLAB), 00290 Helsinki, Finland; 6Department of Veterinary Biosciences, University of Helsinki, 00290 Helsinki, Finland; 7Department of Internal Medicine, Tampere University Hospital, Elämänaukio 2, 33520 Tampere, Finland; satu.makela@pshp.fi (S.M.); jukka.mustonen@tuni.fi (J.M.); 8Faculty of Medicine and Health Technology, Tampere University, 33014 Tampere, Finland

**Keywords:** Puumala hantavirus, acute kidney injury, proteinuria, glycocalyx, podocytes, heparanase, syndecan–1

## Abstract

Old–world orthohantaviruses cause hemorrhagic fever with renal syndrome (HFRS), characterized by acute kidney injury (AKI) with transient proteinuria. It seems plausible that proteinuria during acute HFRS is mediated by the disruption of the glomerular filtration barrier (GFB) due to vascular leakage, a hallmark of orthohantavirus–caused diseases. However, direct infection of endothelial cells by orthohantaviruses does not result in increased endothelial permeability, and alternative explanations for vascular leakage and diminished GFB function are necessary. Vascular integrity is partly dependent on an intact endothelial glycocalyx, which is susceptible to cleavage by heparanase (HPSE). To understand the role of glycocalyx degradation in HFRS–associated proteinuria, we investigated the levels of HPSE in urine and plasma during acute, convalescent and recovery stages of HFRS caused by Puumala orthohantavirus. HPSE levels in urine during acute HFRS were significantly increased and strongly associated with the severity of AKI and other markers of disease severity. Furthermore, increased expression of HPSE was detected in vitro in orthohantavirus–infected podocytes, which line the outer surfaces of glomerular capillaries. Taken together, these findings suggest the local activation of HPSE in the kidneys of orthohantavirus–infected patients with the potential to disrupt the endothelial glycocalyx, leading to increased protein leakage through the GFB, resulting in high amounts of proteinuria.

## 1. Introduction

Orthohantaviruses are zoonotic pathogens able to cause two major clinical syndromes: hemorrhagic fever with renal syndrome (HFRS) and hantavirus cardiopulmonary syndrome (HCPS). Different orthohantavirus species possess varying pathogenicity in humans. The prototype orthohantavirus Hantaan (HTNV, circulates in Eastern Asia) causes a more severe form of HFRS with mortality up to 5%, whereas Puumala orthohantavirus (PUUV, mainly in Northern Europe) causes a mild form of the disease often referred to as nephropathia epidemica (NE, mortality 0.1%). Orthohantaviruses circulating in the Americas are associated with the highly severe HCPS (mortality > 30%) [1]. HFRS is characterized by changes in the endothelial barrier functions, fluid alterations including extravasation, increased glomerular permeability and proteinuria [2,3,4]. 

Renal involvement in acute NE includes transient proteinuria, microscopic hematuria, and oliguric acute kidney injury (AKI). This is followed by a polyuric phase and finally recovery [4]. Proteinuria in NE is mostly composed of albumin, but larger proteins such as IgG are also found in the urine, suggesting proteinuria to have a glomerular origin [5]. Albuminuria makes a flash–like appearance and returns rapidly to normal levels in 2–3 weeks [6], and the amount of this albuminuria on admission to hospital predicts the severity of upcoming AKI [7]. Additionally, a concomitant urinary loss of low–molecular–weight proteins such as β2– and α1–microglobulin indicates that tubular injury also contributes to the proteinuria. A typical renal biopsy finding is acute tubulointerstitial nephritis with only minor glomerular light microscopic abnormalities. The glomerular barrier breakdown seems to be caused by the release of cytokines and other vasoactive factors, rather than by endothelial cell death [4]. By electron microscopy, however, podocyte foot–process effacement has been documented [8]. 

AKI, an acute decline in glomerular filtration rate (GFR) determined by elevated serum creatinine levels, is found in most hospital–treated patients in Finland [4] and presents with a favorable outcome [9].

The functional structure in the kidney preventing the excretion of large plasma proteins into the urine is known as the glomerular filtration barrier (GFB). The GFB is composed of pedicles of podocytes, the glomerular basement membrane, and the glomerular endothelium of capillaries [10]. The active role that the glomerular endothelium plays in renal filtration is linked to its negatively charged surface of proteoglycan glycosaminoglycans (GAG) and glycoprotein molecules: the glycocalyx [11,12,13,14,15]. It acts by preventing negatively charged molecules of similar or larger size than albumin to be excreted through the urine, a process known as charge selectivity [16]. Importantly, the increased glomerular permeability seen in PUUV–HFRS was demonstrated to be associated with impairment of the size– and charge–selectivity properties of the GFB [5]. Interestingly, orthohantaviruses can infect different cells of the human kidney, including glomerular endothelial cells, tubular epithelial cells, and podocytes, resulting in the disruption of their cell–to–cell interactions [17,18]. Furthermore, recent findings suggest podocyte injury to be the main cause of proteinuria in PUUV–HFRS [19].

Damage to the endothelial glycocalyx has been implicated in vascular permeability, one of the major pathophysiological aspects of orthohantaviral diseases [20]. Therefore, it is necessary to study the possibility of the enzymatic destruction of these endothelial components, which has been demonstrated to result in increased urinary excretion of albumin in previous studies [21,22,23,24,25]. 

The function of heparan sulfate (HS), the major GAG component [26,27,28,29], is regulated by active heparanase (HPSE), which cleaves HS on its polymeric chains and internal sites [30]. Studies performed on mice demonstrated the importance of GAGs in renal charge selectivity [31] and the effect of HPSE in degrading the volume fraction of negatively charged fibers in the glycocalyx [22]. Similarly, syndecan–1 is a heparan sulfate proteoglycan (HSPG) present on the surfaces of endothelial cells. In addition to the cleavage of syndecan–1 HS side chains, HPSE also promotes the cleavage and release of its biologically active ectodomain. Thus, syndecan–1 is a renowned marker of glycocalyx degradation [32,33,34]. Moreover, syndecan–1 is known to bind to many mediators of disease pathogenesis [35] and, strikingly, it has been found to be elevated in the circulating blood during PUUV–HFRS, where syndecan–1 levels associated with albumin levels and vascular leakage parameters, as well as disease severity [36].

The aim of this study is to investigate the mechanisms of glycocalyx degradation during PUUV infection by determining the concentration of HPSE in the plasma and urine of PUUV–infected patients, as well as in orthohantavirus–infected human kidney cells, to better understand the underlying physiopathological processes during acute PUUV–HFRS.

## 2. Materials and Methods

### 2.1. Ethics Statement and Clinical Samples

The Ethics Committees of Tampere University Hospital (permit number R04180) approved the use of patient samples. All subjects gave written informed consent in accordance with the Declaration of Helsinki. The study material consisted of plasma and urine from hospitalized, serologically confirmed acute PUUV infection at Tampere University Hospital, Finland, between 2005 and 2009. The samples were collected sequentially during acute (hospitalization) and convalescent (20–30 days after onset of fever) phases, as well as 6 months and one year after full recovery, used as controls. Sequential samples included plasma as well as urine from 56 patients (Table 1). The samples were stored at −80 °C prior to analysis. Urinary creatinine was determined by an Atellica CH5 analyzer (Siemens; Berlin, Germany) at the Helsinki and Uusimaa Hospital District diagnostic laboratory. Daily white blood cell (WBC), plasma C–reactive protein (CRP) and serum creatinine concentrations were determined at the Laboratory Centre of the Pirkanmaa Hospital District (Tampere, Finland) using standard methods. Daily urine creatinine and albumin concentrations were determined at the University of Massachusetts Memorial Health Care Hospital Labs using standard methods [6]. The estimated glomerular filtration rate (eGFR) was calculated using the Chronic Kidney Disease Epidemiology Collaboration (CKD–EPI) equation [37].

The overall severity of patients was assessed by a score system adapted from the sequential organ failure assessment scoring system, where the maximum levels of plasma creatinine (4 = >440, 3 = 300–440, 2 = 171–299, 1 = 110–170 and 0 = <110 μmol/L), minimum level of thrombocytes (4 = <20, 3 = 20–49, 2 = 50–99, 1 = 100–150 and 0 = >150 × 109/L) and lowest mean arterial blood pressure (MAP) measured during hospitalization (1 = <70 and 0 = ≥70 mmHg) were ranked. 

### 2.2. HPSE Measurement Assay

HPSE levels from patient samples and cell culture supernatants were measured through the enzyme’s ability to cleave HS, which was quantified with the use of a standard curve, as described in [38]. In detail, Nunc maxisorp flat–bottom 96–well plates (Thermo scientific; Breda, The Netherlands) were coated with 10 ug/mL heparan sulfate from bovine kidney (HSBK) (Sigma–Aldrich; Zwijndrecht, The Netherlands) in coating buffer (3.3 M ammonium sulfate ((NH_4_)_2_SO_4_)), for 1h at 37 °C. Subsequently, plates were washed with PBS supplemented with 0.05% Tween 20 (PBST) and blocked with 1% BSA in PBS at RT. After blocking, plates were washed with PBST, followed by a final washing step with PBS. Plasma and urine samples were then incubated for 2 h at 37 °C, in a 1:4 dilution in HPSE buffer (50 mM citric acid–sodium citrate, 50 mM NaCl, 1 mM CaCl2 at pH 5.0). Next, plates were washed with PBST and incubated with primary mouse anti–rat IgM HS antibody JM403 (Amsbio; Abingdon, United Kingdom, cat. no. #370730–S, RRID: AB_10890960, 1 μg/mL in PBST) for 1 h at RT, washed with PBST and then incubated with secondary goat anti–mouse IgM HRP antibody (Southern Biotech; Uden, The Netherlands, cat. no. #1020–05, RRID: AB_2794201, 1:10,000 dilution in PBST) for 1h at RT and washed with PBST once again. Finally, 3,3′,5,5′–tetramethylbenzidine (TMB) substrate (Sigma–Aldrich, Zwijndrecht, The Netherlands) was added to the plates, the reaction was stopped by the addition of 0.5 M sulfuric acid, and absorbance was measured at 450 nm. The HPSE concentration in the plasma and urine of patient samples, as well as from podocyte cultures, was compared to a standard curve of recombinant human HPSE (Bio–techne; Abingdon, UK, Cat#7570–GH–005). Urine concentrations measured were then normalized by dividing the values by creatinine (heparanase:creatinine ratio).

### 2.3. ELISAs

The syndecan–1 ELISA kit was purchased from R&D Systems and used according to the manufacturer’s protocol (Human Syndecan–1 DuoSet, Catalog no. DY2780; Bio–techne; Abingdon, UK). Cytokines interleukin (IL)–6, IL–8, monocyte chemoattractant protein (MCP)–1 and interferon–gamma induced protein (IP)–10 were measured previously from plasma and urine by Luminex [39]. Myeloperoxidase (MPO) was measured from plasma previously [40]. The concentrations measured in urine were normalized by dividing the values by creatinine (syndecan–1:creatinine ratio, IL–6:creatinine ratio, IL–8:creatinine ratio, IP–10:creatinine ratio, MCP–1:creatinine ratio).

### 2.4. Virus Isolates

The PUUV–Suonenjoki (PUUV–Suo) strain was propagated in a bank vole renal epithelial cell line (MyGlaRec.B from EVAg), grown in Dulbecco’s minimum essential medium—high glucose (DMEM; Sigma Aldrich; Zwijndrecht, The Netherlands) supplemented with 10% inactivated FCS, 100 IU/mL penicillin, 100 μg/mL streptomycin, 2 mM L–glutamine and a mix of non–essential amino acids (Sigma Aldrich; Zwijndrecht, The Netherlands). The HTNV strain 76–118 was grown in Vero E6 cells (green monkey kidney epithelial cell line; ATCC no. CRL–1586) in Minimum Essential Medium (MEM; Sigma–Aldrich; Zwijndrecht, The Netherlands) and supplemented with 10% inactivated FCS, 100 IU/mL of penicillin and 100 µg/mL of streptomycin and 2 mM of L–glutamine. Viruses were purified from cell culture supernatants by ultracentrifugation through a 30% sucrose cushion (SW28 rotor, 27,000 rpm, 50 min, +4 °C) and suspended to the corresponding growth medium. The infectious titers of PUUV and HTNV stocks were routinely 10^5^ and 10^7^ focus–forming units (FFU)/mL, respectively. Virus titers were measured by incubating diluted virus stocks with Vero E6 cells for 24 h at 37 °C, followed by acetone fixation and staining with a rabbit polyclonal antibody specific for PUUV nucleocapsid (N) protein (anti–PUUN) and AlexaFluor488–conjugated donkey anti–rabbit secondary antibody (Thermo Scientific; Breda, The Netherlands). Fluorescent focus–forming units (FFFU)/mL were counted under a UV microscope (Zeiss Axio Imager 1; Zeiss, Jena, Germany). Where indicated, viruses were inactivated using UV crosslinker (300,000 μJ/cm^2^, Stratalinker, Stratagene).

### 2.5. Podocyte Cultures

Conditionally immortalized human podocytes (AB 8/13) [41] were cultured in RPMI–1640 growth medium supplemented with 10% fetal calf serum (FCS) and 1% ITS (Sigma–Aldrich; Zwijndrecht, The Netherlands), maintained at 33 °C and shifted to 37 °C for differentiation, on Viewplate black 96–well plates (PerkinElmer, Inc; Waltham, MA, USA). At 8 days of differentiation, podocyte cultures were infected with UV–inactivated or live virus isolates at the indicated multiplicity of infection (MOI) for 1 h at 37 °C, after which the virus–containing medium was changed back to podocyte growth medium. At indicated days post–infection (dpi), supernatants were collected and frozen in −20 °C and cells were washed with PBS prior to fixation with 4% paraformaldehyde (PFA) for 10 min. 

### 2.6. Immunofluorescence

The podocyte infection frequency was assessed by staining permeabilized (PBS supplemented with 3% BSA and 0.1% TritonX–100 for 10 min) cells with anti–PUUN rabbit serum followed by AlexaFluor488–conjugated secondary antibody (Thermo Scientific; Breda, The Netherlands) supplemented with 1:5000 diluted Hoechst 33420. Finally, imaging was performed with a PerkinElmer Opera Phenix (PerkinElmer, Inc; Waltham, MA, USA) spinning disk confocal microscope using a 5× water–immersion objective (NA 1.0). The analysis of the number of infected cells was conducted with the Harmony software (PerkinElmer, Inc; Waltham, MA, USA) by using a supervised linear classifier. 

### 2.7. HPSE mRNA Expression

The expression of HPSE mRNA was determined from RNA isolated from podocytes (*n* = 2 for each infection group and time point) using Trizol according to the manufacturer’s instructions (Thermo Scientific; Breda, The Netherlands). The expression of HPSE and GAPDH mRNAs was quantified by one–step reverse–transcriptase quantitative polymerase chain reaction (RT–qPCR) with commercial FAM– and VIC–based fluorescent primer–probe sets (Thermo Scientific; Breda, The Netherlands), respectively, employing 1–step fast virus master mix (Thermo Scientific; Breda, The Netherlands) and AriaMx real–time PCR instrumentation (Agilent, Santa Clara, CA, USA). The relative expression of HPSE mRNA was calculated by the 2^–deltadeltaCt method [42] using GAPDH for normalization and the average of mock samples (*n* = 4) as a reference.

### 2.8. Statistical Analysis

Statistical analysis was performed using GraphPad Prism 8.3 software (GraphPad Software; San Diego, CA, USA), R software v3.6.3 (R core team) and SPSS v25 (IBM; Armonk, NY, USA). Statistically significant differences between groups were assessed with Kruskal–Wallis or Tukey’s multiple comparisons test, depending on the sample distribution and the number of groups analyzed. Statistically significant correlations between normally and non–normally distributed variables were assessed by calculating Spearman’s correlation coefficients. Following rank transformation of non–normally distributed variables, normality was assessed with Anderson–Darling, D’Agostino–Pearson omnibus, Shapiro–Wilk and Kolmogorov–Smirnov tests. Then, the correlations between variables were examined using the Pearson’s correlation coefficient test to confirm that multicollinearity between variables did not exist, to conduct simple and linear regression analyses. Statistically significant differences during various time points after the onset of fever (aof), using the last convalescence time point (365 days aof) as the reference category, were assessed by generalized estimating equations (GEE), assuming a linear distribution and setting the working correlation matrix to independent.

## 3. Results

### 3.1. HPSE and Syndecan–1 Levels Are Increased in Urine of Acute PUUV–HFRS Patients

In support of potential glycocalyx degradation during acute PUUV–HFRS, HPSE and syndecan–1 concentrations were significantly increased in urine samples collected from patients during the acute (days aof 3–9) phase of PUUV–HFRS, compared to the convalescence (days aof 20–30) and control phase (182 and 365 aof) (Figure 1A,B). The elevated levels of urinary HPSE and syndecan–1 coincided with increased levels of albumin in urine during the acute stage of the disease (Figure 1C). However, plasma HPSE in the acute phase was not elevated, as compared to the control phase (Figure 1D). Taken together, these findings suggest that if HPSE affects endothelial glycocalyx degradation during PUUV–HFRS, its effect is exerted locally in the kidneys, rather than systemically. 

### 3.2. Urinary HPSE Levels Correlate with Albuminuria and Other Disease Severity Markers

Next, we considered whether there was a link between HPSE, syndecan–1 and albumin measured from urine, as well as their association with different inflammatory cytokines and chemokines measured previously from the same set of patients, and clinical parameters describing the severity of this disease. Significant bivariate associations were assessed by calculating Spearman’s rank correlation coefficients (Figure 2), and urinary HPSE showed a significant positive correlation with urinary albumin levels (*p <* 0.0001, r = 0.734), followed by overall disease severity evaluated with the severity score (*p =* 0.007, r = 0.394). Additionally, HPSE in urine was significantly higher in patients with a lower eGFR, the hallmark of AKI (*p =* 0.046, r = −0.299). Albuminuria did, however, show a significant positive correlation with urinary interleukin 6 (*p =* 0.049, r^2^ = 0.340), an inflammatory cytokine directly leaking from plasma to urine, as well as with total blood leukocyte counts (*p =* 0.015, r^2^ = 0.368) and granulocyte–derived myeloperoxidase (MPO) levels in circulating blood (*p =* 0.010, r^2^ = 0.381). In contrast, urinary HPSE did not show significant correlations with these parameters. Thus, the lack of association between urinary HPSE and markers of systemic inflammation points towards the local expression of HPSE in the kidneys of acute PUUV–HFRS. As for urinary syndecan–1, its levels displayed a significant positive correlation with the pro–inflammatory plasma IL–6 (*p =* 0.011, r^2^ = 0.439), IL–8 (*p =* 0.002, r^2^ = 0.512) and MCP–1 (*p =* 0.010, r^2^ = 0.44) levels. In addition to the correlations shown in Figure 2, in which urine values were normalized with urinary creatinine levels, correlations using the absolute urinary concentrations were also assessed (Appendix A). The significant associations of urinary HPSE and syndecan–1 with other parameters tested were essentially the same and generally with increased *p*–values.

### 3.3. Urinary HPSE Levels Possess Predictive Power over Disease Severity, Albuminuria and Hypotension 

Following the assessment of significant correlations between different parameters, we wished to investigate the possible predictive power between urinary HPSE and other variables of interest. For this purpose, we initially performed single regression models of urinary albumin and urinary HPSE levels as predictors of disease severity, assessed with a severity score, and found both to explain severity to various degrees: urinary albumin levels could explain 19% of the variability in disease severity (*p =* 0.0025, r^2^ = 0.1934); meanwhile, urinary HPSE could account for 13% of the variability in the severity score (*p =* 0.0132, r = 0.1346) (Figure 3A,B, Table 2).

To demonstrate the link between plasma albumin and the vasculature, we assessed the association between this circulating protein and the minimum mean arterial pressure (MAP) presented by patients during their hospital stay. Results exhibited a correlation between low MAP and a decreased plasma albumin concentration (*p =* 0.003, r^2^ = 0.492) (Figure 2). Moreover, a linear regression model between these two variables confirmed their strong association (*p =* 0.0031, r^2^ = 0.2425) (Figure 3C). Additionally, combining plasma albumin with other variables of relevance (HPSE in urine or plasma, albumin in urine and eGFR) in multiple regression analysis models revealed that the predictive power of plasma albumin on hypotension was strengthened only in the presence of both urinary HPSE and urinary albumin levels (*p =* 0.0064, r^2^ = 0.3417) (Figure 3D,E), explaining up to 34% of the variability in blood pressure.

Following this discovery, we wished to assess the association between urinary HPSE and other parameters. We found, through simple linear regression analyses, that urinary HPSE was significantly linked to albuminuria (*p <* 0.0001, r^2^ = 0.5381) (Figure 3F), but none of the other independent variables. Although eGFR was not significantly associated with urinary HPSE in a simple regression model, a multiple regression model showed that eGFR combined with urinary HPSE increased the predictive power on albuminuria, explaining up to 64% of its variability (*p <* 0.0001, r^2^ = 0.638) (Figure 3G,H). However, eGFR did not increase the predictive power of urinary albumin on urinary HPSE levels to the same extent (*p <* 0.0001, r^2^ = 0.5773) (not shown), accounting for 58% of the variability in urinary HPSE levels. Furthermore, removal of urinary HPSE from the multiple regression model produced a major drop in the variability prediction and significance to 30% (*p =* 0.0063, r^2^ = 0.3039) (not shown). 

### 3.4. Upregulation of HPSE in Orthohantavirus–Infected Podocytes

Orthohantaviruses have been shown to infect podocytes [17,18], which have been postulated as major contributors of HPSE during kidney diseases (e.g., diabetic nephropathy) [43,44]. To assess the ability of orthohantavirus infection to induce HPSE upregulation in podocytes, we infected conditionally immortalized and differentiated human podocytes with various amounts (high and low dose; MOI 10 and 1, respectively) of Hantaan orthohantavirus (HTNV). We analyzed virus infectivity and HPSE expression by comparing HTNV–infected cells with UV–inactivated HTNV (MOI 10) or mock–infected cells at 2 and 4 dpi (Figure 4). As judged by viral nucleocapsid N–specific intracellular immunofluorescence, we observed efficient infection only in cells infected with a high dose of HTNV (Figure 4A, ~70% of infected cells at both 2 and 4 dpi). Interestingly, in line with infection efficiency, we observed a significant increase in HPSE expression in podocyte supernatants infected with a higher viral dose, but not in those infected with a lower dose or UV–inactivated HTNV (Figure 4B). Consistently, we observed mildly but significantly increased expression of HPSE mRNA only in cells infected with high–dose HTNV (Figure 4C, ~1.5–fold over mock–infected cells), indicating that the increase in HPSE levels is most likely due to increased transcriptional activity.

## 4. Discussion

One of the major pathophysiological aspects of old–world orthohantaviral diseases is the presence of vascular leakage, an increased glomerular permeability linked to size– and charge–selectivity in the glomerular filtration barrier (GFB) and proteinuria. Since the disruption of the endothelial glycocalyx has been implicated as one of the potential mechanisms leading to vascular permeability, this study indirectly assessed endothelial damage through the measurement of the glycocalyx–degrading enzyme HPSE; and a well–known marker of glycocalyx degradation, syndecan–1, from the urine of PUUV–infected patients. Results showed significantly increased HPSE and syndecan–1 protein levels in patient urine during acute PUUV–caused HFRS compared to full recovery, used as a control. Additionally, these parameters significantly correlated with several important inflammatory markers describing disease severity and the extent of AKI: urinary syndecan–1 was associated with indicators of systemic inflammation (plasma IL–6, IL–8 and MCP–1), whereas urinary HPSE was linked to indicators of renal dysfunction (albuminuria, lower eGFR and higher plasma creatinine levels) and finally higher severity scores, representing a more severe overall clinical presentation. This association between urinary HPSE and disease severity was confirmed with a simple regression linear model, where HPSE explained up to 13% of the variability of the severity scores. 

Syndecan–1 has been found to be elevated in the plasma of acute PUUV–HFRS in a previous study [36], and our results exhibited higher syndecan–1 levels in urine during acute disease. Interestingly, though, urinary syndecan–1 did not correlate with the same urinary parameters as HPSE. This difference could be due to its low molecular weight (30 kDa), which makes syndecan–1 more likely to be freely filtered through the glomerulus. While convalescent and recovery control phase samples presented significantly lower levels of syndecan–1 than during the acute infection, syndecan–1 was still detectable in control samples a year after the onset of disease (Figure 1A). It is possible that the epithelial cells of the urinary tract could also produce syndecan–1 into urine, as is the case for Tamm Horsfall protein [45]. Another possibility is a partial tubular reabsorption defect of syndecan–1, possibly resulting from a permanent tubular damage, as seen in some patients presenting with urinary long–lasting α1–microglobulin levels after acute PUUV–HFRS infection [46,47]. HPSE, on the other hand, is an enzyme with a molecular weight of 50 kDa, not detected in control urine. Therefore, it is most likely not freely filtered nor reabsorbed. Thus, it is similar to albumin, the main plasma protein with a high molecular weight (66.5 kDa), known to bypass the GFB only under pathological conditions. 

Altogether, these findings suggest that increased urinary HPSE and syndecan–1 are related to the damage of the renal structures and point towards a degradation of the endothelial glycocalyx during HFRS. This is supported by previous studies demonstrating HPSE as a player in kidney damage and dysfunction, including other renal pathologies that present proteinuria [48,49,50]. 

Our data showed that patients with lower plasma albumin levels presented significantly more severe hypotension during their hospital stay (minimum levels of MAP) (*p =* 0.0031, r = 0.2425), with plasma albumin explaining 24% of the variability in blood pressure. Moreover, the combination of plasma albumin concentration with urinary HPSE and albumin in multiple regression analysis models increased the predictability of hypotension by 10%. Additionally, the subtraction of urinary HPSE from the models resulted in no change in the predictive power of plasma albumin in MAP, together with a decrease in the statistical power (*p =* 0.013 and r^2^ = 0.24 in both cases), compared to the simple regression model from Figure 3D. 

Subsequently, the association between urinary HPSE and albuminuria was modeled in a simple linear regression analysis. This model was significant and displayed urine HPSE explaining up to 54% of the variability of urine albumin. When combined with eGFR, the predictability of urinary HPSE in albuminuria increased to 64%. Overall, the presence of urinary HPSE was required for increasing the predictive power of the multiple regressions modeled for MAP and albuminuria as dependent variables, and important clinical manifestations during HFRS.

Perhaps unexpectedly, the plasma HPSE concentration measured during acute PUUV–HFRS was not increased, but rather seemed somewhat decreased compared to control phase samples (Figure 1D). This can be explained by the leakage of this enzyme into the urine. However, the increased HPSE levels measured from the urine samples do not seem to depend only on this leakage, since plasma HPSE was found to positively correlate with urinary HPSE, albumin and protein. If plasma was the sole source of HPSE in urine, these two variables would present a negative correlation. Thus, these findings suggest an increased HPSE production and not mere relocalization between bodily fluids. In addition, since our HPSE assay measured the active enzyme concentration of HPSE, it is possible that the measured levels of plasma HPSE during acute HFRS were affected by the presence of circulating HPSE inhibitors such as free HS or proteins with HS side chains. It is known that circulating syndecan–1 levels are increased during acute PUUV–HFRS [36]. 

We hypothesized that the increased urinary HPSE levels in PUUV–HFRS could be due to a local increase in the enzyme by the direct viral infection of renal cells. We selected podocytes as our cell culture model due to their known susceptibility to orthohantaviral infection [18,51] and possible causative link to proteinuria in PUUV–HFRS [19], as well as their propensity to increase HPSE production in response to proteinuria–promoting factors such as glucose [43,44]. As our orthohantavirus model virus, we used HTNV, which causes more severe HFRS as compared to PUUV and is more readily grown to high titers in cell culture. We observed that a high initial dose of HTNV (MOI 10) was able to induce increased HPSE levels and mRNA expression in podocytes, which was due to its ability to efficiently replicate (UV–inactivated virus did not show the same effect) in podocytes during the timeframe set up for our experiment. With a lower infectious dose of HTNV (MOI 1), no infection was observed during the 4 dpi, which is in line with the slow infection kinetics of orthohantaviruses in podocytes [18,51,52]. We also carried out PUUV infections in podocytes, but, due to the relatively low titers achieved for PUUV in cell culture, we were not able to replicate the high initial virus dose of HTNV and therefore could not detect sufficient infection by PUUV in our assay setup. The molecular mechanisms leading to increased expression of HPSE mRNA in infected podocytes are unclear but could involve increased expression of pro–inflammatory cytokines such as tumor necrosis factor α, which is known to induce HPSE [53] and is upregulated in acute PUUV–HFRS [54].

Our data indicate that an efficient old–world orthohantavirus infection of podocytes can result in increased HPSE secretion, which is likely to contribute to elevated HPSE levels in patient urine, possibly promoting endothelial cell permeability and proteinuria locally in patient kidneys by degrading endothelial glycocalyx HS chains, subsequently altering the GFB’s charge–selectivity. Whether increased HPSE levels, and thus activity, in infected podocytes could contribute to decreased podocyte motility and the disruption of cell–to–cell contacts, as observed by other investigators [18,51], remains to be determined. The extensive crosstalk between podocytes and glomerular endothelial cells is well known [55] and endothelial glycocalyx degradation would likely have adverse effects also on podocyte functions and could potentially lead to podocyte injury. 

One of the limitations of this study was that syndecan–1 was not measured from plasma, since the employed ELISA assay did not show the required specificity in the plasma samples used for this study. In addition, although eGFR is accurate to assess kidney function during chronic kidney injury, its use during AKI is controversial because of its fast onset and resolution. However, we decided to use eGFR instead of plasma creatinine as an indication of AKI in this study, since it takes the gender and weight of the patients into account. Another clinical parameter describing disease severity in this study was minimum MAP. Hypotonic shock is the most common cause of death in the more severe forms of HFRS, but the patients included in this study did not suffer from hemodynamic failure or hypotonia and only 5 out of 56 patients had minimum MAP below 70 mmHg during hospital stay. This indicates that our patient cohort had a relatively mild course of HFRS, which is typical for PUUV infections. Thus, while low MAP is not necessarily important for the pathogenesis of PUUV–caused HFRS, we feel that its prominence in the more severe forms of the disease justifies its inclusion as one of the clinical parameters in our analysis. This can hopefully indirectly provide insight into the possible role of the currently measured parameters, HPSE and syndecan–1, also in more severe HFRS. Furthermore, it needs to be added that podocytes are not necessarily the only source of HPSE in the kidneys during acute HFRS, and infected glomerular endothelial cells could also contribute to the overall HPSE levels.

## 5. Conclusions

Acute PUUV–HFRS presented with elevation of HPSE and syndecan–1 levels in urine. These urinary HPSE levels were strongly associated with overall disease severity and albuminuria. Moreover, multiple regression analyses revealed that urinary HPSE increases the predictive power on dependent variables such as MAP and urinary albumin. Finally, we demonstrated through in vitro assays that HTNV infection of podocytes led to the upregulation of HPSE. Therefore, HPSE activity is potentially upregulated during HFRS, which could be the cause of renal endothelium glycocalyx degradation. Further studies are needed on this topic, but these results point towards the potential use of HPSE inhibitors as therapeutic treatment options during acute HFRS.

## Figures and Tables

**Figure 1 viruses-14-00450-f001:**
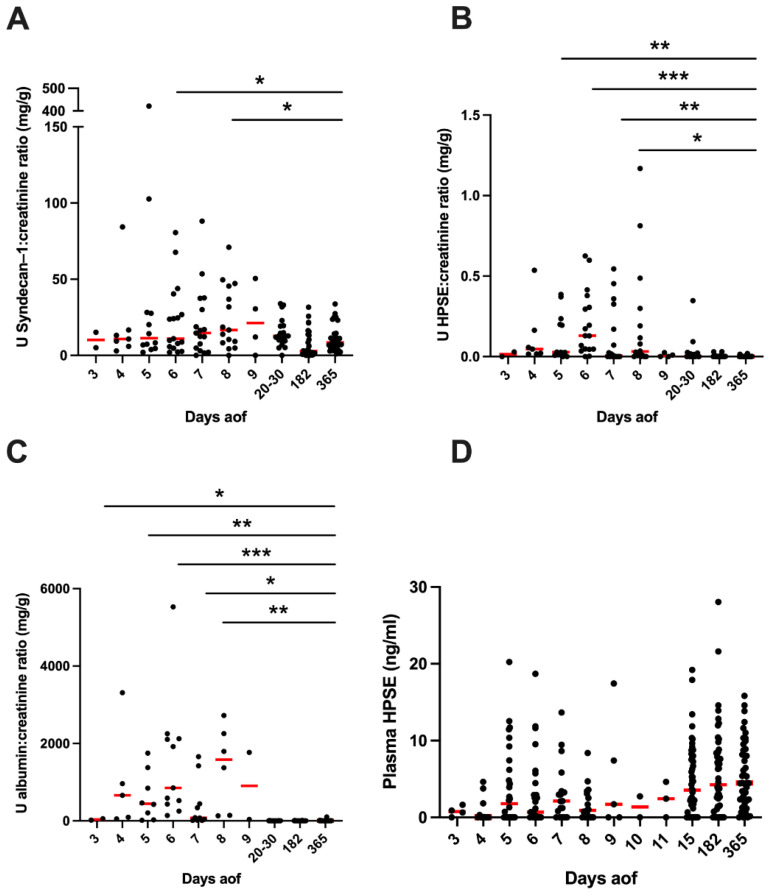
Heparanase and syndecan–1 levels during PUUV–HFRS. (**A**) Urinary syndecan–1:creatinine ratio (**B**), HPSE:creatinine ratio and (**C**) albumin:creatinine ratio. (**D**) HPSE concentration in plasma. Concentrations were measured from sequential samples of 49 (55 for plasma) patients hospitalized due to PUUV–HFRS (with a total of 159 urine and 262 plasma samples). Days after onset of fever (aof) 3–9 represent the acute stage, 20–30 the convalescent and 182–365 the controls. Differences between individual time points and the last time point (365 aof), which was considered to represent full recovery and used as controls, were assessed by GEE and significant differences indicated as *** <0.001, ** <0.01 and * < 0.05. Red lines represent mean ± standard deviation.

**Figure 2 viruses-14-00450-f002:**
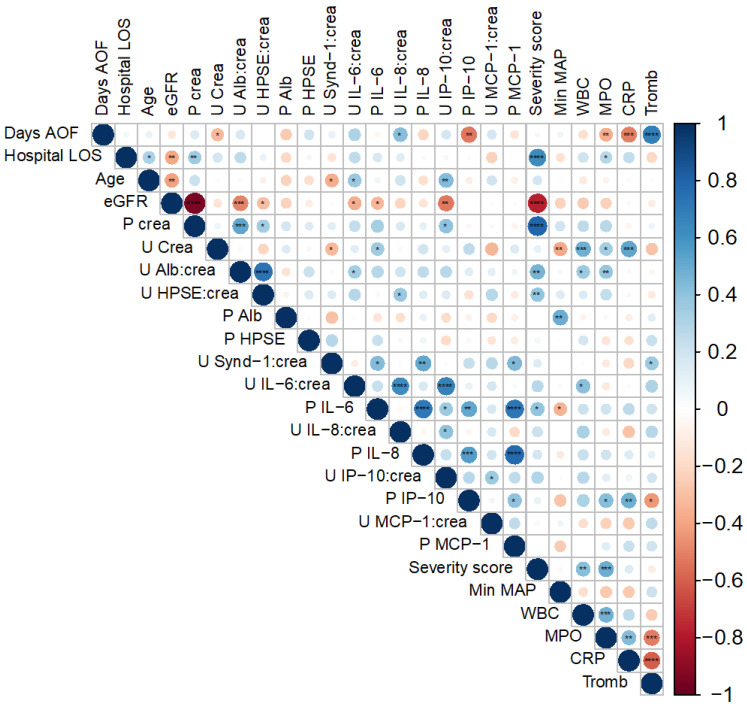
Spearman’s rank correlation coefficient matrix. The colors of the round circles indicate the level of the correlation coefficient and increasing size with a lower *p*–value. Statistical significance is specified as * = *p* < 0.05, **= *p* < 0.01, ***= *p* < 0.001, ****= *p* < 0.0001. AOF = after onset of fever, LOS = length of stay, eGFR = estimated glomerular filtration rate, P = plasma, U = urine, crea = creatinine, alb = albumin, HPSE = heparanase, Synd–1 = syndecan–1, IL = interleukin, IP–10 = interferon gamma–induced protein 10, MCP–1 = monocyte chemoattractant protein 1, min MAP = minimum mean arterial pressure, WBC = white blood cells, MPO = plasma myeloperoxidase, CRP = plasma C–reactive protein, tromb = blood thrombocytes.

**Figure 3 viruses-14-00450-f003:**
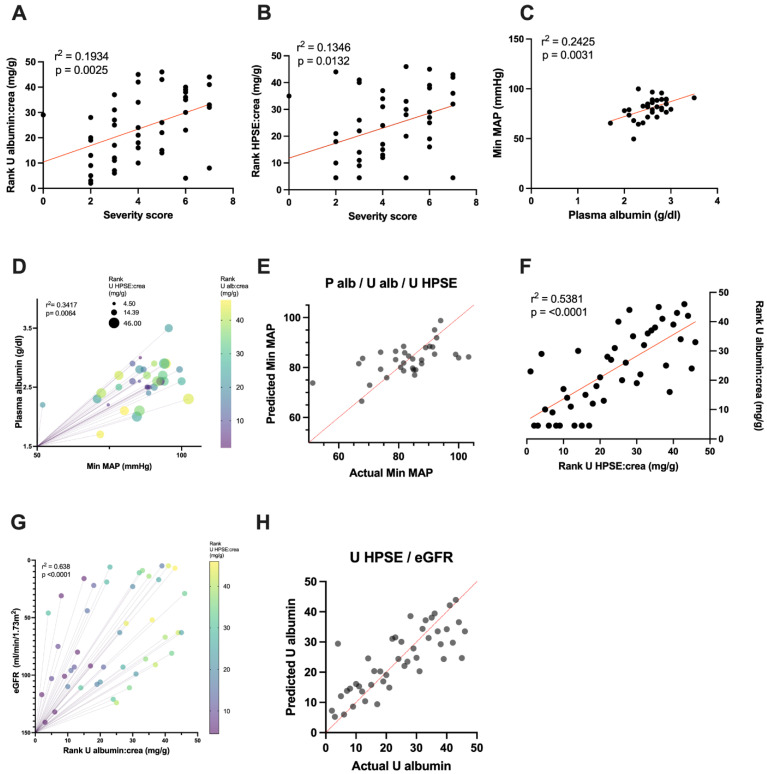
Simple and multiple linear regression models. Simple linear regression models of rank–transformed urinary albumin:creatinine ratio (**A**) and rank–transformed urinary HPSE:creatinine ratio (**B**) with severity score. (**C**) Simple linear regression model of plasma albumin and MAP. (**D**) Multiple linear regression model of plasma albumin, rank–transformed urinary HPSE:creatinine ratio and rank–transformed albumin on MAP. (**E**) Predicted vs. actual MAP of multiple linear regression model in (**D**). (**F**) Simple linear regression model of rank–transformed urinary HPSE:creatinine ratio and albumin:creatinine ratio. (**G**) Multiple linear regression model of eGFR and urinary HPSE on albuminuria. (**H**) Predicted vs. actual data of the multiple regression model in (**G**).

**Figure 4 viruses-14-00450-f004:**
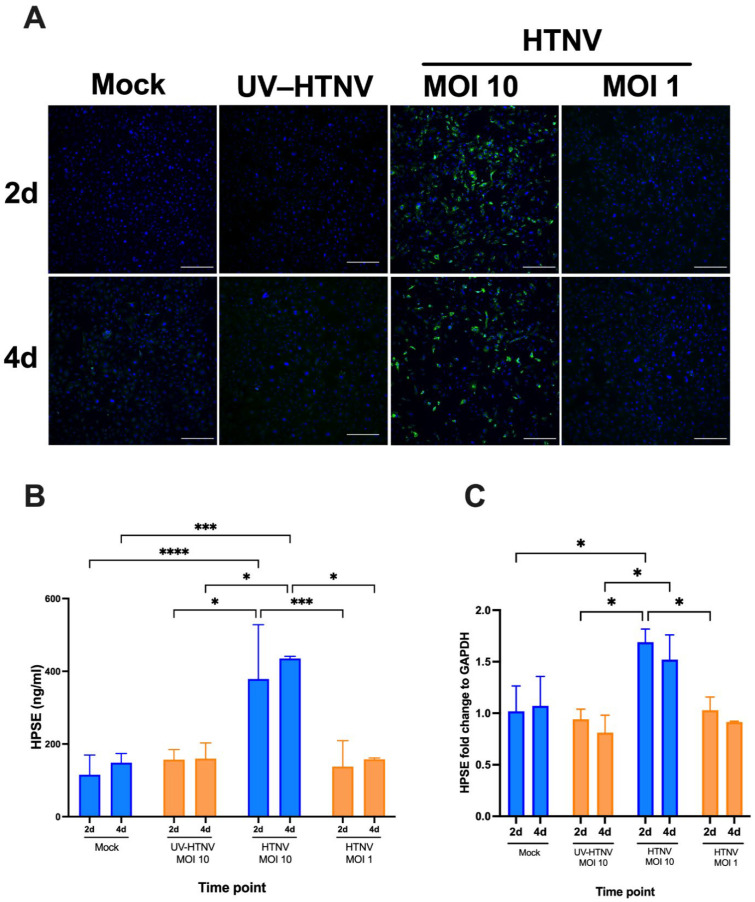
Increased HPSE concentration in HTNV–infected podocytes. Podocytes were infected with live HTNV (MOI 10 or 1), UV–inactivated HTNV (MOI 10) or remained uninfected (mock). Supernatants were collected and cells either fixed for immunofluorescence analysis or subjected to RNA extraction at 2 and 4 dpi. (**A**) Fixed and permeabilized podocytes were stained with rabbit serum against viral nucleocapsid N protein followed by AF488–conjugated secondary antibody (green) or Hoechst 33,420 to detect nuclei (blue). (**B**) HPSE levels measured from podocyte supernatants. (**C**) Isolated RNA was subjected to multiplex RT–qPCR with primers and probes detecting HPSE and GAPDH mRNA. The signal for HPSE mRNA was normalized based on GAPDH mRNA levels and fold change calculated in reference to mock–infected podocytes. Statistically significant differences were assessed with Tukey’s multiple comparisons test. ****, *** and * indicate *p* < 0.0001, *p* < 0.001 and *p* < 0.05, respectively.

**Table 1 viruses-14-00450-t001:** Clinical and laboratory parameters of PUUV–HFRS patients. Abbreviations: eGFR = estimated glomerular filtration rate, HPSE = heparanase, IL = interleukin, IP–10 = interferon gamma–induced protein–10, MCP–1 = monocyte chemoattractant protein–1, MPO = myeloperoxidase.

Variable (Unit)	Mean (SE; Range)
Number of patients	56
Hospital length of stay (days)	5.81 (0.46; 2–25)
Age (years)	41 (1.65; 22–73)
Male: Female ratio	2.3:1
Min eGFR (mL/min/1.73m^2^)	63 (5.75; 5–141)
Max plasma creatinine (μmol/L)	266.22 (35.61; 45–1071)
Urine albumin: creatinine ratio (mg/g)	1012.12 (190.67; 2.9–5530.8)
Max urine HPSE: creatinine ratio (mg/g)	0.22 (0.04; 0–1.17)
Min plasma albumin (g/dL)	2.41 (0.08; 2.3–3.5)
Max plasma HPSE (ng/mL)	4.54 (0.70; 0–20.23)
Max urine syndecan–1: creatinine ratio	36.3 (9.5; 0–411.62)
Urine IL–6: creatinine ratio (pg/μg)	0.21 (0.05; 0.005–1.30)
Plasma IL–6 (pg/mL)	128.58 (39.9; 10–785.8)
Urine IL–8: creatinine ratio (pg/μg)	0.45 (0.18; 0.003–5.80)
Plasma IL–8 (pg/mL)	41.26 (8.72; 10–234.86)
Urine IP–10: creatinine ratio (pg/μg)	4.47 (0.70; 0.22–16.92)
Plasma IP–10 (pg/mL)	4734.39 (764.17; 739.8–18224)
Urine MCP–1: creatinine ratio (pg/μg)	7.64 (1.97; 0.01–63.64)
Plasma MCP–1 (pg/mL)	373.97 (120.78; 23.5–3636.78)
MPO (ng/mL)	150.85 (9.5; 46–332.17)
Severity score	4.04 (0.24; 0–7)
Min mean arterial pressure (mmHg)	84.11 (1.42; 51.33–103.33)
Max CRP (mg/L)	67.32 (5.11; 9.7–155.2)
Min thrombocytes (10^9^/L)	85.56 (5.66; 36–266)
Max leukocytes (10^9^/L)	10.62 (0.58; 4.1–25.5)

**Table 2 viruses-14-00450-t002:** Simple and multiple regression models.

Dependent Variables	Type of Regression	Independent Variables	Estimate	Std. Error	95% CI	t Value	*p*–Value of Variable	R^2^	F– Statistic	*p*–Value of Model
Minimum mean arterial pressure	Simple	Plasma albumin	15.594	4.715	[5.488–24.7]	3.440	0.0031 **	0.2425	10.25	0.0031 **
	Multiple	Plasma albumin	15.4735	4.8731	[5.507–25.44]	3.175	0.0035 **	0.3417	5.017	0.0064 **
		Urinary albumin: creatinine ratio	−0.1508	0.1774	[−0.5136–0.2121]	−0.85	0.0724			
		Urinary HPSE: creatinine ratio	0.3703	0.1854	[−0.0088–0.7495]	1.998	0.0502			
Severity score	Simple	Urinary albumin: creatinine ratio	3.254	1.014	[1.210–5.299]	3.211	0.0025 **	0.1934	10.31	0.0025 **
	Simple	Urinary HPSE: creatinine ratio	2.799	1.082	[0.6167–4.982]	2.586	0.0132 *	0.1346	6.690	0.0132 *
Urinary albumin: creatinine ratio	Simple	Urinary HPSE: creatinine ratio	0.7355	0.1027	[0.5257–0.9376]	0.1027	<0.0001 ***	0.5276	51.26	<0.0001 ***
	Multiple	Urinary HPSE: creatinine ratio	0.6587	0.0941	[0.4687–0.8486]	6.998	<0.0001 ***	0.621	37.02	<0.0001 ***
		eGFR	−0.0848	0.0308	[−0.1469–−0.0227]	−2.756	0.0086 **			

Statistical significance is specified as * = *p* < 0.05, ** = *p* < 0.01, *** = *p* < 0.001.

## Data Availability

Not applicable.

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
