# Peer review of "Increased Heparanase Levels in Urine during Acute Puumala Orthohantavirus Infection Are Associated with Disease Severity"

_viruses, 2022, doi:10.3390/v14030450_

Round 1

Reviewer 1 Report

Cabrera et al aim to study the mechanisms leading to AKI in Puumala hantavirus infection. They investigate the potential role of heparanase activity in disruption of glomerular filtration barrier.  The aim to analyze 1) human patient population and 2) podocyte cell culture model.

Diverse roles of heparanase in AKI is reviewd by Abassi (Adv Exp Med Biol 2020).  They consider the complex biology and clinical significance of heparanase on kidneys. In summary, several carefully regulated activation steps and inhibitory mechanisms are needed to control heparanase activity. It must be appreciated that urine haparanase can be elevated in various conditions including acute and chronic kidney diseases, malignancies, infections, ischemia and inflammation. Summary of heparanase functions according to Uniprot database include: “inactive at neutral pH but becomes active under acidic conditions such as during tumor invasion and in inflammatory processes. Facilitates cell migration associated with metastasis, wound healing and inflammation. Enhances shedding of syndecans, and increases endothelial invasion and angiogenesis in myelomas. Acts as procoagulant by increasing the generation of activation factor X in the presence of tissue factor and activation factor VII. Increases cell adhesion to the extracellular matrix (ECM), independent of its enzymatic activity. Induces AKT1/PKB phosphorylation via lipid rafts increasing cell mobility and invasion. Heparin increases this AKT1/PKB activation. Regulates osteogenesis. Enhances angiogenesis through up-regulation of SRC-mediated activation of VEGF. Implicated in hair follicle inner root sheath differentiation and hair homeostasis.”

While the question presented by the authors is interesting, the complexity of AKI development in HFRS and diversity of heparanase biology is obvious. Selected concerns include:

  1. Any marker of inflammatory and tissue injury can be high during acute infection with hantaviruses or other infectious agents. Proof of causality between the heparanase and AKI is a very difficult question. Investigations in heparanase deficient vs wild type animal models – if those are even available for hantavirus studies – should possibly be more informative? Urine heparanase and other markers of this study can be elevated only because of unspecific tissue injury and leakage?
  2. Selected markers of disease severity in this study may be misleading. SOFA score has been mainly developed for truly septic intensive care unit patient (ICU) population. ICU treatment and mechanical ventilation with hemodynamic failure is extremely rare in Puumalavirus infection. Thrombocytopenia and creatinine alone (without signs of true multiorgan failure) can produce high SOFA scores. This evaluation is not appropriate.
  3. Another marker – the lowest measured mean arterial pressure – is also misleading in Puumalavirus HFRS. The patients can be hypotonic at the time of hospital entry. However, as mentioned above, a true hemodynamic failure and lasting or progressive hypotonia caused by Puumalavirus is not common. Mean length of hospitalization was 6 days suggesting that the study population represented mild HFRS.
  4. Figure 1 demonstrates that timing of elevated U syndecan and U heparanase was similar to that of U albumin. These elevated values are observed during oliguric phase of the disease? Total amount of heparanase and syndecan is not elevated when the low amount of urine is considered? Figure 1D demonstrates that plasma heparanse is low at the time (days 3 to 4) of hospital entry?
  5. In summary (points 1 to 4), I ask the authors to be critical: 1) they need to further show that the used patient modelling is appropriate to draw the current conclusions 2) SOFA and mean arterial pressure parameters need critical consideration 3) make sure that the observed results are not caused by oliguria.
  6. In vitro podocyte modelling supports the findings. However, infection itself can cause many unspecific changes to transcription and eventually lead to cell death. The authors should provide more evidence on how specific the elevation in heparanase mRNA is? Any speculation on mechanisms on effects of the virus on heparanase gene regulation?

My conclusion is that while the experiments are carefully conducted and the data is solid, the research question is very difficult. The authors should reconsider 1) how they present the results 2) what conclusions can be made: what has truly been shown by the used methods and what remains speculative? 3) are they able to provide more data to support their hypothesis?

Author Response

Cabrera et al aim to study the mechanisms leading to AKI in Puumala hantavirus infection. They investigate the potential role of heparanase activity in disruption of glomerular filtration barrier.  The aim to analyze 1) human patient population and 2) podocyte cell culture model.

Diverse roles of heparanase in AKI is reviewd by Abassi (Adv Exp Med Biol 2020).  They consider the complex biology and clinical significance of heparanase on kidneys. In summary, several carefully regulated activation steps and inhibitory mechanisms are needed to control heparanase activity. It must be appreciated that urine haparanase can be elevated in various conditions including acute and chronic kidney diseases, malignancies, infections, ischemia and inflammation. Summary of heparanase functions according to Uniprot database include: “inactive at neutral pH but becomes active under acidic conditions such as during tumor invasion and in inflammatory processes. Facilitates cell migration associated with metastasis, wound healing and inflammation. Enhances shedding of syndecans, and increases endothelial invasion and angiogenesis in myelomas. Acts as procoagulant by increasing the generation of activation factor X in the presence of tissue factor and activation factor VII. Increases cell adhesion to the extracellular matrix (ECM), independent of its enzymatic activity. Induces AKT1/PKB phosphorylation via lipid rafts increasing cell mobility and invasion. Heparin increases this AKT1/PKB activation. Regulates osteogenesis. Enhances angiogenesis through up-regulation of SRC-mediated activation of VEGF. Implicated in hair follicle inner root sheath differentiation and hair homeostasis.”

While the question presented by the authors is interesting, the complexity of AKI development in HFRS and diversity of heparanase biology is obvious. Selected concerns include:

Any marker of inflammatory and tissue injury can be high during acute infection with hantaviruses or other infectious agents. Proof of causality between the heparanase and AKI is a very difficult question. Investigations in heparanase deficient vs wild type animal models – if those are even available for hantavirus studies – should possibly be more informative? Urine heparanase and other markers of this study can be elevated only because of unspecific tissue injury and leakage?

RESPONSE: We agree with the reviewer regarding the difficulties to show causality just by measuring human patient samples, which typically lack samples taken at disease onset. Since there are currently no animal models for HFRS, we need to rely on in vitro cell culture models to study the mechanistic details of HFRS. Therefore, without the possibility to study HPSE in the kidneys of patients/animals with acute HFRS, we instead investigated HPSE activity in podocyte cultures infected with an orthohantavirus and believe the results are supportive of our conclusions.

Selected markers of disease severity in this study may be misleading. SOFA score has been mainly developed for truly septic intensive care unit patient (ICU) population. ICU treatment and mechanical ventilation with hemodynamic failure is extremely rare in Puumalavirus infection. Thrombocytopenia and creatinine alone (without signs of true multiorgan failure) can produce high SOFA scores. This evaluation is not appropriate.

RESPONSE: We agree that the actual SOFA scoring system is intended for patients at ICU and we have only applied the scoring system for HFRS (therefore we named it adapted SOFA; aSOFA). However, we have renamed our scoring system to “severity score” in the revised MS to avoid confusion with the actual SOFA. 

Another marker – the lowest measured mean arterial pressure – is also misleading in Puumalavirus HFRS. The patients can be hypotonic at the time of hospital entry. However, as mentioned above, a true hemodynamic failure and lasting or progressive hypotonia caused by Puumalavirus is not common. Mean length of hospitalization was 6 days suggesting that the study population represented mild HFRS.

RESPONSE: We agree that the patients did not suffer from hemodynamic failure or hypotonia and only 5 out of 56 patients had minimum mean arterial pressure below 70 mmHg during hospital stay. This certainly indicates that compared to HFRS caused by many other orthohantaviruses our patient cohort had a relatively mild course of HFRS, which is typical for Puumalavirus infections. However, since hypotonic shock is the most common cause of death in the more severe forms of HFRS, we think it is still interesting to keep minimum mean arterial pressure as one of the clinical parameters in our analysis. This can hopefully provide insight on the possible role of currently measured parameters HPSE and syndecan-1 also in the case of the more severe forms HFRS.

Figure 1 demonstrates that timing of elevated U syndecan and U heparanase was similar to that of U albumin. These elevated values are observed during oliguric phase of the disease? Total amount of heparanase and syndecan is not elevated when the low amount of urine is considered? Figure 1D demonstrates that plasma heparanse is low at the time (days 3 to 4) of hospital entry?

RESPONSE: Oliguria is typically associated with acute HFRS. However, we did not find evidence of oliguria in our patient cohort at the time of sampling. The median urine output during the acute stage samples was 1.2 ml/min and at 1-year follow up sampling 1.0 ml/min. It is likely that most of the patients have already passed the oliguric phase at the time of sampling. Based on this, it is evident that the urine concentrations of both HPSE and syndecan-1 are elevated due to excess production (or diminished glomerular filtration barrier) in urine during acute Puumala-caused HFRS.   

In summary (points 1 to 4), I ask the authors to be critical: 1) they need to further show that the used patient modelling is appropriate to draw the current conclusions 2) SOFA and mean arterial pressure parameters need critical consideration 3) make sure that the observed results are not caused by oliguria.

RESPONSE: Please see above responses to the critical comments.

In vitro podocyte modelling supports the findings. However, infection itself can cause many unspecific changes to transcription and eventually lead to cell death. The authors should provide more evidence on how specific the elevation in heparanase mRNA is? Any speculation on mechanisms on effects of the virus on heparanase gene regulation?

RESPONSE: Our data show that elevation of HPSE mRNA in podocytes is dependent on virus replication. This can be concluded from the finding that UV-inactivated replication-incompetent virus did not induce HPSE. As the reviewer points out, it is possible that the induction of HPSE by replicating virus is indirect (i.e. mediated by host cell factors). In fact, HPSE transcription can be induced by pro-inflammatory factors such as TNF-alpha. We added the following to the discussion:

rows 467-471: “The molecular mechanisms leading to increased expression of HPSE mRNA in infected podocytes are unclear but could involve increased expression of pro-inflammatory cytokines such as tumor necrosis factor α, which is known to induce HPSE [54] and is upregulated in acute PUUV-HFRS [55].

My conclusion is that while the experiments are carefully conducted and the data is solid, the research question is very difficult. The authors should reconsider 1) how they present the results 2) what conclusions can be made: what has truly been shown by the used methods and what remains speculative? 3) are they able to provide more data to support their hypothesis?

RESPONSE: It is true that our hypothesis would benefit from additional in vivo data. The best way to further explore our hypothesis would be through a disease animal model for HFRS, which unfortunately is still lacking. Based on reviewer criticism we have revised the title of our MS, which doesn’t anymore state increased HPSE levels in the kidneys but only in the urine. The revised title is “Increased  heparanase levels in urine during acute Puumala orthohantavirus infection is associated with disease severity”

Reviewer 2 Report

Increased heparanase activity in the kidneys during acute Puumala orthohantavirus infection

I read with great interest the authors' data regarding potential degradation of endothelial glycocalyx during acute HFRS. Since a disruption of the glomerular filtration barrier (GFB) is a hallmark of acute hantavirus infection, the authors address an important and insufficiently understood problem. A better understanding of the pathogenesis is urgently needed. By investigating longitudinal urine levels of heparanase in 56 patients, the authors show an association of urinary heparanase (and in part syndecan-1) and the severity of AKI as well as with other severity markers such as albuminuria and certain cytokines in blood and urine during the acute phase of hantavirus infection. In addition, an increased expression of heparanase was observed in vitro in hantavirus-infected podocytes as a potential source of local heparanase. The authors conclude that these findings suggest a local activation of heparanase in the kidneys leading to a disruption of the endothelial glycocalyx thereby impairing the GFB with subsequent proteinuria. Although this hypothesis would represent a new pathophysiological feature of great interest, the authors need to clarify several issues to support the conclusions presented. As mentioned in the discussion section, I have concerns that these findings may be explained by a simple loss of heparanase and others via the impaired GFB rather than representing a true pathophysiological link. Therefore, the following issues need to be addressed to further support the importance of the presented data:

Major comments:

  • Normalization of urine parameters by calculating a ratio to urinary creatine is a proven method to account for different urine volumes in patients with impaired but stable renal function and to improve comparability (e.g. albumin-to-creatinine ratio). However, in patients with acute renal failure, urinary creatinine excretion may decrease dramatically or fluctuate widely over time (due to progressive impairment of kidney function) leading to an artificial increase in the calculated ratios. To exclude the possibility that reduced urinary creatinine excretion is the main driver of the observed increase in the various calculated ratios of several urinary parameters, the authors should report both normalized and absolute concentrations for all analyses (at least in the supplemental materials)
    • It also remains unclear why the authors normalized some of the urinary parameters such as syndecan-1, HPSE, and albumin with creatinine, but not others such as urinary cytokines. Since the same theoretical principles apply to all urinary biomarkers, this mixed analysis is rather unusual if a relevant association needs to be demonstrated and limits comparability. I suggest performing correlation analyses of absolute and normalized values separately
  • My main concern is that the presented findings may be simply explained by leakage of the investigated enzymes/molecules into the urine due to the general and well known impairment of the GFB during acute hantavirus infection. Lower plasma levels of HPSE especially in the acute phase (where GFR impairment is present) support this theory. In this regard, I would also be interested in syndecan-1 levels in the blood of these patients. The authors state that increased urinary HPSE levels do not seem to depend only on leakage, since plasma HPSE was found to positively correlate with urinary HPSE, albumin and protein and not negatively. However, the rate of production of HPSE in blood might exceed the fraction that enters urine and explain why no negative correlation is shown. The authors should comment on this. To further support their theory, the authors should also provide correlation analyses with absolute urinary HPSE levels (not normalized to creatinine). It might be useful to show some exemplary biomarker courses of individual patients in order to better illustrate the correlations (absolute and normalized).
  • The authors should clarify and discuss more in detail HPSE levels/production rate in blood and urine of healthy individuals compared to acute hantavirus infection. I also suggest including a healthy cohort in the manuscript as a control.
  • Since it was shown that podocyte injury contributes to impaired GFB integrity, the authors should comment/discuss how their findings/degradation of glycocalyx fit into this context.
  • I would be also interested when the lowest MAP appeared. Was it during acute phase or in the polyuric phase? How many patients had a clinically relevant MAP decrease? Since MAP is used as a severity marker, it is important to provide evidence that the extent of MAP decline was clinically relevant.

Minor comments:

  • The title is a slightly misleading since the authors did not study heparanase activity in the kidneys, but "only" in vivo and in urine and blood.
  • Page 6, 224-225: for better understanding, the authors should clarify that the talk about HPSE in blood.

Author Response

I read with great interest the authors' data regarding potential degradation of endothelial glycocalyx during acute HFRS. Since a disruption of the glomerular filtration barrier (GFB) is a hallmark of acute hantavirus infection, the authors address an important and insufficiently understood problem. A better understanding of the pathogenesis is urgently needed. By investigating longitudinal urine levels of heparanase in 56 patients, the authors show an association of urinary heparanase (and in part syndecan-1) and the severity of AKI as well as with other severity markers such as albuminuria and certain cytokines in blood and urine during the acute phase of hantavirus infection. In addition, an increased expression of heparanase was observed in vitro in hantavirus-infected podocytes as a potential source of local heparanase. The authors conclude that these findings suggest a local activation of heparanase in the kidneys leading to a disruption of the endothelial glycocalyx thereby impairing the GFB with subsequent proteinuria. Although this hypothesis would represent a new pathophysiological feature of great interest, the authors need to clarify several issues to support the conclusions presented. As mentioned in the discussion section, I have concerns that these findings may be explained by a simple loss of heparanase and others via the impaired GFB rather than representing a true pathophysiological link. Therefore, the following issues need to be addressed to further support the importance of the presented data:

Major comments:

Normalization of urine parameters by calculating a ratio to urinary creatine is a proven method to account for different urine volumes in patients with impaired but stable renal function and to improve comparability (e.g. albumin-to-creatinine ratio). However, in patients with acute renal failure, urinary creatinine excretion may decrease dramatically or fluctuate widely over time (due to progressive impairment of kidney function) leading to an artificial increase in the calculated ratios. To exclude the possibility that reduced urinary creatinine excretion is the main driver of the observed increase in the various calculated ratios of several urinary parameters, the authors should report both normalized and absolute concentrations for all analyses (at least in the supplemental materials)

RESPONSE: The reviewer is raising a valid point regarding fluctuation of urinary creatinine levels during an acute kidney disease. We are now providing a similar correlogram as in Fig. 2 but using the absolute concentrations instead normalized values (supplementary Fig.1). The main conclusions remain the same although we generally observe slightly increased significances between  urinary HPSE and syndecan—1 with other parameters. We decided to keep the correlogram with normalized values in the main MS and absolute values as supplemental data. 

It also remains unclear why the authors normalized some of the urinary parameters such as syndecan-1, HPSE, and albumin with creatinine, but not others such as urinary cytokines. Since the same theoretical principles apply to all urinary biomarkers, this mixed analysis is rather unusual if a relevant association needs to be demonstrated and limits comparability. I suggest performing correlation analyses of absolute and normalized values separately

RESPONSE: Good point by the reviewer. We had made a mistake in Table 1 where we showed only the absolute concentrations of the cytokines. However, normalized values were used in the correlogram. We made the appropriate corrections in table1 and now provide correlograms with both normalized and absolute values in the revised MS.  

My main concern is that the presented findings may be simply explained by leakage of the investigated enzymes/molecules into the urine due to the general and well known impairment of the GFB during acute hantavirus infection. Lower plasma levels of HPSE especially in the acute phase (where GFR impairment is present) support this theory. In this regard, I would also be interested in syndecan-1 levels in the blood of these patients. The authors state that increased urinary HPSE levels do not seem to depend only on leakage, since plasma HPSE was found to positively correlate with urinary HPSE, albumin and protein and not negatively. However, the rate of production of HPSE in blood might exceed the fraction that enters urine and explain why no negative correlation is shown. The authors should comment on this. To further support their theory, the authors should also provide correlation analyses with absolute urinary HPSE levels (not normalized to creatinine). It might be useful to show some exemplary biomarker courses of individual patients in order to better illustrate the correlations (absolute and normalized).

RESPONSE: The theory that the increase in HPSE (and syndecan-1) in urine is due to diminished GFB is plausible and probably explains part of the observed urinary increase of these factors. However, if this would be the only reason, you would expect a negative or no correlation between plasma and urinary HPSE. However, a positive correlation between these compartments suggests that there is a general increase of HPSE during acute HFRS.

In fact, in addition to leakage, the diminished active concentration of HPSE in acute plasma can also be explained by increased presence of HPSE inhibitors. We added the following to the discussion.

rows 446-451: “In addition, since our HPSE assay measures the active enzyme concentration of HPSE, it is possible that the measured levels of plasma HPSE during acute HFRS are affected by the presence of circulating HPSE inhibitors such as free HS or proteins with HS-side chains. It is known that circulating syndecan-1 levels are increased during acute PUUV-HFRS [37].”

As already mentioned in the discussion, we also tried to measure syndecan-1 in plasma with ELISA but unfortunately the assay resulted in unspecific background from the human plasma samples and we could not therefore reliably measure this factor in our patient blood samples.

The authors should clarify and discuss more in detail HPSE levels/production rate in blood and urine of healthy individuals compared to acute hantavirus infection. I also suggest including a healthy cohort in the manuscript as a control.

RESPONSE: We consider that the 6- 12- month follow up samples obtained for most of the patients in our cohort represent fully recovery. Using follow up samples from the same patients is probably the most appropriate control group for the acute stage samples in terms of age, sex and different kinds of non-identifiable individual clinical determinants. Based on this data, the active concentration of HPSE in urine of recovered healthy individuals is negligible while it is readily measurable from blood.  This point has been addressed in the discussion.

Since it was shown that podocyte injury contributes to impaired GFB integrity, the authors should comment/discuss how their findings/degradation of glycocalyx fit into this context.

RESPONSE: It is possible that glycocalyx degradation and ensuing vascular permeability contributes to podocyte injury by allowing molecules that are toxic to podocytes to flow through the first line of the GFB. We added the following to the discussion:

Rows 479-482: “The extensive cross-talk between podocytes and glomerular endothelial cells is well-known [54] and endothelial glycocalyx degradation would likely have adverse effects also on podocyte functions and could potentially lead to podocyte injury.”

I would be also interested when the lowest MAP appeared. Was it during acute phase or in the polyuric phase? How many patients had a clinically relevant MAP decrease? Since MAP is used as a severity marker, it is important to provide evidence that the extent of MAP decline was clinically relevant.

RESPONSE: Lowest MAP was typically seen at admission. The patients did not suffer from hemodynamic failure or hypotonia and only 5 out of 56 patients had minimum mean arterial pressure below 70 mmHg during hospital stay. This certainly indicates that compared to HFRS caused by many other orthohantaviruses our patient cohort had a relatively mild course of HFRS, which is typical for Puumalavirus infections. However, since hypotonic shock is the most common cause of death in the more severe forms of HFRS, we think it is still interesting to keep minimum mean arterial pressure as one of the clinical parameters in our analysis. This can hopefully provide insight of the possible role of currently measured parameters HPSE and syndecan-1 also in the case of the more severe forms HFRS.

Minor comments:

The title is a slightly misleading since the authors did not study heparanase activity in the kidneys, but "only" in vivo and in urine and blood.

RESPONSE: The title has been changed to “Increased heparanase levels in urine during acute Puumala orthohantavirus infection is associated with disease severity”.

Page 6, 224-225: for better understanding, the authors should clarify that the talk about HPSE in blood.

RESPONSE: The text in that paragraph has been modified, specifying better when the HPSE measurements were either from urine or from plasma.

Reviewer 3 Report

The manuscript presented by Cabrera and coauthors describes an increased activity of heparanase in urinary samples of HFRS patients and in in vitro infected podocytes. The work provides interesting and novel findings concerning the pathogenesis of acute kidney injury in HFRS and disease severity. Interestingly, the authors used a human podocyte cell line. The use of an adequate cell culture system in virological studies is noteworthy, because most researchers perform their experiments in non-target or even in non-human cells and cell lines.

Some points need clarification before publication: heparanase activity is analyzed (title and method section); however, in text, tables, axis titles the authors state that heparanase levels are increased. What was analyzed in the samples? Activity or protein levels of heparanase?

Is the effect specific for infected podocytes? If not, the possibility that other glomerular cell types also exhibit enhanced heparanase activity upon infection should be mentioned in the discussion.

In addition, some minor points should be improved: In table 1 the last row is empty and should be removed. The term „live HTNV“ (figure 4A and its legend) is not really correct. Viruses do not live. I would prefer „replication competent virus“ , „non-inactivated virus“ or simply „HTNV“.

Author Response

The manuscript presented by Cabrera and coauthors describes an increased activity of heparanase in urinary samples of HFRS patients and in in vitro infected podocytes. The work provides interesting and novel findings concerning the pathogenesis of acute kidney injury in HFRS and disease severity. Interestingly, the authors used a human podocyte cell line. The use of an adequate cell culture system in virological studies is noteworthy, because most researchers perform their experiments in non-target or even in non-human cells and cell lines.

Some points need clarification before publication: heparanase activity is analyzed (title and method section); however, in text, tables, axis titles the authors state that heparanase levels are increased. What was analyzed in the samples? Activity or protein levels of heparanase?

RESPONSE: HPSE levels from patient samples and cell culture supernatants were measured through the enzyme’s ability to cleave HS, which was quantified with the use of a standard curve. This has been better specified in the methods section.

Is the effect specific for infected podocytes? If not, the possibility that other glomerular cell types also exhibit enhanced heparanase activity upon infection should be mentioned in the discussion.

RESPONSE: Yes, for instance glomerular endothelial cells could potentially produce HPSE and since these cells are likely infected during the course of the disease, would present a likely additional source of HPSE. We added the following in the discussion:

Rows 489-491: “Furthermore, it needs to be added that podocytes are not necessarily the only source of HPSE in the kidneys and infected glomerular endothelial cells could also contribute to the overall HPSE levels during acute HFRS.”

In addition, some minor points should be improved: In table 1 the last row is empty and should be removed. The term „live HTNV“ (figure 4A and its legend) is not really correct. Viruses do not live. I would prefer „replication competent virus“ , „non-inactivated virus“ or simply „HTNV“.

RESPONSE: The figure text has been adapted to simply HTNV.

Round 2

Reviewer 1 Report

They authors have have responded to all major concerns.

Author Response

Thank you!

Reviewer 2 Report

Great work! Pleasure to read!

As a final point, I would suggest including the authors' response to "MAP and disease severity"  (last major comment) in the limitations section, including the rationale for why this analysis is still important.

Author Response

Thank you!

We added the following to the end of discussion (rows 473-482):

"Another clinical parameter describing disease severity in this study was minimum MAP. Hypotonic shock is the most common cause of death in the more severe forms of HFRS, but the patients included in this study did not suffer from hemodynamic failure or hypotonia and only 5 out of 56 patients had minimum MAP below 70 mmHg during hospital stay. This indicates that our patient cohort had a relatively mild course of HFRS, which is typical for PUUV infections. Thus, while low MAP is not necessarily important for the pathogenesis of PUUV-caused HFRS, we feel that its prominence in the more severe forms of the disease justifies its inclusion as one of the clinical parameters also in our analysis. This can hopefully indirectly provide insight of the possible role of currently measured parameters HPSE and syndecan-1 also in more severe HFRS.